# Expression of Rejection-Associated Transcripts in Early Protocol Renal Transplant Biopsies Is Associated with Tacrolimus Exposure and Graft Outcome

**DOI:** 10.3390/ijms25063189

**Published:** 2024-03-10

**Authors:** Betty Chamoun, Irina B. Torres, Alejandra Gabaldón, Thomas Jouvé, María Meneghini, José M. Zúñiga, Joana Sellarés, Manel Perelló, Daniel Serón, Oriol Bestard, Francesc Moreso

**Affiliations:** 1Nephrology Department, Hospital Universitari Vall d’Hebron, 08035 Barcelona, Spain; md.chamounbetty@gmail.com (B.C.); maria.meneghini@vallhebron.cat (M.M.); josemiguel.zuniga@vallhebron.cat (J.M.Z.); joana.sellares@vallhebron.cat (J.S.); manel.perello@vallhebron.cat (M.P.); danielseron56@gmail.com (D.S.); oriol.bestard@vallhebron.cat (O.B.); 2Department of Medicine, Autonomous University of Barcelona, 08035 Barcelona, Spain; 3Laboratory of Nephrology and Transplantation, Vall d’Hebron Institut de Recerca (VHIR), 08035 Barcelona, Spain; tjouve@chu-grenoble.fr; 4Pathology Department, Hospital Universitari Vall d’Hebron, 08035 Barcelona, Spain; mariaalejandra.gabaldon@vallhebron.cat; 5Renal Transplant Unit, Nephrology Department, Grenoble University Hospital, 38043 Grenoble, France

**Keywords:** renal transplantation, biopsies, rejection, gene expression, tacrolimus

## Abstract

Subclinical inflammation in protocol biopsies relates to tacrolimus exposure and human leukocyte antigen (HLA) matching. We aimed to characterize transcripts associated with rejection and tacrolimus exposure and the latter’s association with transplant outcomes. We tested whether gene expression is associated with rejection using strictly normal protocol biopsies (n = 17) and biopsies with T cell-mediated rejection (TCMR) or antibody-mediated rejection (ABMR) according to Banff criteria (n = 12). Subsequently, we analyzed these transcripts in a set of 4-month protocol biopsies (n = 137) to assess their association with donor and recipient characteristics, the intensity of immunosuppression, and the graft outcome. Differential expression (false discovery rate (FDR) < 0.01, fold (change (FC) > 3) between normal and rejection biopsies yielded a set of 111 genes. In the protocol biopsy cohort (n = 137), 19 out of these 111 genes correlated with tacrolimus trough levels at the time of biopsy (TAC-C_0_), and unsupervised analysis split this cohort into two clusters. The two clusters differed in donor age and tacrolimus trough levels. Subclinical rejection, including borderline lesions, tended to occur in the same cluster. Logistic regression analysis indicated that TAC-C_0_ at the time of biopsy (OR: 0.83, 95%CI:0.72–0.06, *p* = 0.0117) was associated with cluster 2. In a follow-up averaging 70 ± 30 months, this patient group displayed a significant decline in renal function (*p* = 0.0135). The expression of rejection-associated transcripts in early protocol biopsies is associated with tacrolimus exposure and a faster decline in renal function.

## 1. Introduction

Routine graft monitoring in renal transplantation relies on non-invasive biomarkers such as serum creatinine, proteinuria, and HLA antibodies. More than 30 years ago, to explore the feasibility of histological monitoring, some centers started programs of protocol biopsies and observed that there were grafts with stable function depicting histological changes of rejection, leading to the definition of subclinical rejection (SCR) [1,2,3]. During the cyclosporine era, SCR was prevalent (>30%) and it was proven that its treatment better preserves renal function [2,4]. However, a clinical trial addressing SCR treatment in patients on modern immunosuppression with tacrolimus/MMF yielded limited clinical benefit due to the low rate of SCR [5]. For this reason, it has also been explored whether minor histological changes are associated with graft outcome. In a study including a large set of 6-month protocol biopsies (n = 957), it was shown that interstitial inflammation (i-score > 0) in otherwise normal protocol biopsies is associated with a significantly lower 15 y graft survival, comparable to SCR or interstitial fibrosis/tubular atrophy (IF/TA) with inflammation [6].

Although the evaluation of renal biopsies based on the Banff classification for renal transplant pathology has been refined since 1991 [7], some uncertainties persist, notably concerning the presence of borderline changes suspicious for T cell-mediated rejection (TCMR) and the incomplete phenotypes of antibody-mediated rejection (ABMR). To further characterize underlying mechanisms leading to different histological phenotypes, an analysis of the transcriptome has been incorporated [8,9,10,11]. It has been shown that molecular diagnostics allow the detection of transcript sets strongly associated with TCMR and have proven useful in differentiating borderline infiltrates likely to lead to the development of overt TCMR and/or graft fibrosis [12]. Recently, we have shown that a rejection-associated gene expression score is present in 83% of protocol biopsies with SCR but only in 17% of protocol biopsies with borderline changes [13]. Importantly, to distill information from RNA microarrays that evaluate thousands of genes, the Banff group has delineated gene sets related to TCMR, ABMR, tissue-repair injury, and other pathways implicated in graft dysfunction [9,10,11,12,13,14].

In studies of serial protocol biopsies performed in renal transplants, the prevalence of SCR is maximal during the initial three months, progressively decreases until the first year, and persists in a small number of patients after the first year. Risk factors associated with SCR are the number of human leukocyte antigen mismatches, the degree of sensitization, retransplantation, the presence of previous clinical acute rejection episodes, and the immunosuppressive regimen being lower in patients treated with tacrolimus and MMF [3]. Our group and others have shown that reduced exposure to tacrolimus and/or MMF is associated with a higher incidence of subclinical inflammation in protocol biopsies performed during the first year [15,16]. Furthermore, while certain studies have linked SCR to HLA ABDR allelic mismatch [17], disparities at the molecular level might offer more informative insights [18]. Notably, in liver transplant recipients, reduced immunosuppression exposure and an increased number of HLA epitope mismatches between donor and recipient have been implicated in the molecular pathogenesis of subclinical liver allograft damage driven by an interferon gamma-orchestrated cellular immune response [19].

In this current study, we employ microfluidic cards to scrutinize the transcriptome of a predefined set of genes related to different histological diagnoses (mainly TCMR and ABMR) previously described by the Banff group [14]. We aim to determine whether transcripts increase or decrease in biopsies with rejection, reflecting changes in resident and/or infiltrating cells. To achieve this, we compare gene expression normal-protocol biopsies and for-cause biopsies that meet the Banff criteria for TCMR or ABMR. Subsequently, we quest these transcripts in a large set of early protocol biopsies to evaluate whether gene expression is associated with donor and recipient characteristics, including the intensity of immunosuppression and donor–recipient HLA mismatch at the allelic or molecular level.

## 2. Results

### 2.1. Patients and Biopsies

In the present study, we have included three groups of patients: patients with a strictly normal early protocol biopsy (group I, n = 17); patients with a biopsy for cause displaying rejection, either TCMR or ABMR, (group II, n = 12); and a large cohort of patients with a protocol biopsy displaying different histological phenotypes (group III, n = 137). Donor and recipient characteristics as well as transplant-related variables from the three studied groups are shown in Table 1. Biopsies were evaluated according to the 2019 Banff criteria [9]. The timing of the biopsy and laboratory data at the time of the biopsy are detailed in Table 2. In the rejection group II (n = 12), there was a mix of cases with TCMR (n = 5), active ABMR (n = 5), and mixed rejection (n = 2). In group III, which contained only protocol biopsies (n = 137), the histological Banff categories were as follows: non-specific changes (n = 40), subclinical TCMR (n = 5), subclinical ABMR (n = 3), borderline changes (n = 16), interstitial fibrosis and tubular atrophy (IF/TA) without interstitial inflammation (n = 59) and IF/TA with interstitial inflammation (IF/TA + i) (n = 14). These results agreed with the prevalence of the different histological phenotypes in the cohort of early protocol biopsies (n = 397) obtained at our center (40.1%, 3.1%, 2.7%, 11.5%, 34.3%, and 8.3%, respectively).

To evaluate whether there was an association between subclinical inflammation and donor/recipient characteristics, transplant-related variables, or immunosuppression, we compare patients in whom the protocol biopsy showed interstitial inflammation (i > 0) and patients in whom the protocol biopsy did not show interstitial infiltrates (i = 0). Among the evaluated variables, subclinical inflammation was associated with prolonged cold ischemia time (*p* = 0.040) and lower tacrolimus trough levels (TAC-C_0_) at the time of biopsy (*p* = 0.002) (Table 3 and Table 4).

Donor and recipient demographics, HLA mismatches at the allelic or molecular level, the presence of delayed graft function, the timing of the biopsy, and renal function did not differ between groups (Table 3). At the time of biopsy, three out of four patients with DSA displayed subclinical ABMR. Multivariate logistic regression analysis showed that TAC-C_0_ (odds ratio [OR]: 0.76; 95% confidence interval [CI]: 0.63–0.92; *p*-value = 0.004) was associated with i-score > 0 while cold ischemia time was on the verge of significance (OR: 1.06; 95% CI:.0.99–1.13; *p*-value = 0.077).

### 2.2. Transcriptome Analysis by Microfluidics

The gene expression in the three groups of biopsies was firstly analyzed by principal component analysis (PCA), and it can be observed that biopsies from group I (normal) and biopsies from group II (TCMR/ABMR) cluster in different areas of the plot while the largest sample of protocol biopsies (group III) clusters in between (Figure 1). The most relevant genes in PCA were ADAMDEC1, CCL5, CLEC4C, CXCL13, and CXCL9 for component 1; and COL1A1, NPHS1, NPHS2, SLC22A2, and SLC4A1 for component 2. As expected, the gene expression comparison between group I and group II (adjusted *p*-value < 0.01 and fold change > 3) yielded as many as 111 differentially expressed genes (Appendix A). These 111 genes extracted from the list provided by the Banff group [14] were mainly related to TCMR (72 genes), ABMR (16 genes), and tissue damage (8 genes).

### 2.3. Transcriptome Analysis and Clinical Variables

We evaluated whether there was an association between TAC-C_0_ at the time of biopsy and the 111 genes associated with rejection (Appendix A). In group I (strictly normal protocol biopsies) there was a close correlation between TAC-C_0_ at the time of biopsy and expression of IKZF3 and CD2 genes (Figure 2). Conversely, in group II (biopsies for cause with TCMR/ABMR) there was no correlation between TAC-C_0_ at the time of biopsy and the expression of any gene. Finally, 19 genes mainly related to TCMR (12 out of 19) correlated with TAC-C_0_ in group III.

Unsupervised cluster analysis allowed two clusters of biopsies to be defined, one containing all normal protocol biopsies (with one exception) and the other containing all biopsies with rejection (Figure 3).

Protocol biopsies from Group III were distributed in a similar proportion in both clusters (77 in Cluster 1 and 60 in Cluster 2). We compared both clusters of biopsies from group III and observed that older donors and lower tacrolimus trough levels at the time of biopsy were grouped in cluster 2 (Table 5).

Noticeably, subclinical rejection including borderline lesions (19 out of 24 cases) and biopsies with IF/TA + i (8 out of 14 cases) also tended to be grouped in cluster 2 (Figure 4). Logistic regression analysis showed that only TAC-C_0_ at the time of biopsy (OR: 0.83, 95% CI: 0.72–0.96, *p*-value = 0.0117) was associated with cluster 2.

### 2.4. Renal Outcome and Protocol Biopsies

At a mean follow-up of 70 ± 30 months, the mean decline in renal function was −0.9 ± 3.9 mL/min/1.73 m^2^/year in the group of patients with a protocol biopsy (group III). While the mean annual decline in renal function did not differ significantly between patients with (i > 0) or without inflammation (−1.1 ± 2.9 vs. −0.8 ± 4.3 mL/min/1.73 m^2^/year, Table 4), it was significantly higher in patients from cluster 2 than in patients from cluster 1 (−1.9 ± 4.1 vs. −0.2 ± 3.7 mL/min/1.73 m^2^/year; *p* = 0.0135; Table 5). Moreover, this difference was independent of the presence of inflammation in the protocol biopsy (cluster 1 with i = 0 (n = 66) −0.20 ± 3.9; cluster 1 with i > 0 (n = 11) +0.19 ± 1.99; *p* = 0.691; and cluster 2 with i = 0 (n = 33) −2.03 ± 4.8; cluster 2 with i > 0 (n = 27) −1.71 ± 3.23; *p* = 0.763).

The total number of patients developing de novo DSA during follow-up in our cohort was low (n = 4; 2.9%). Kaplan–Meier analysis showed that patients with and without inflammation have a non-different rate of development of de novo DSA (*p* = 0.876), while this rate tended to be higher in patients from cluster 2 (*p* = 0.073).

## 3. Discussion

We conducted a prospective study on a set of 4-month protocol biopsies to evaluate whether validated rejection-associated transcripts are associated with tacrolimus exposure at the time of biopsy. The main findings of our study are that we confirm the discrimination capacity between normal and rejection biopsies of a large set of these genes and that the expression of 19 rejection-associated transcripts in early protocol biopsies is associated with tacrolimus exposure at the time of biopsy. Cluster analysis using this set of 19 genes identified a pool of patients with a higher proportion of inflammatory phenotypes, including TCMR, borderline lesions, and IFTA with inflammation. Interestingly, patients from this cluster had less exposure to tacrolimus and displayed a faster decline in renal function during follow-up. The low rate of de novo DSA development in our cohort (2.9%) limits further analysis of its association with subclinical inflammation. Thus, our results suggest that adjusted immunosuppression during the early months after transplantation favors a better control of the inflammatory response without deleterious effects on renal function in the mid-term.

Tacrolimus is the mainstay of immunosuppressive regimens for kidney transplantation since it prevents T cell activation and proliferation. Although tacrolimus reduces the acute rejection rate and improves short-term outcomes after kidney transplantation, it is associated with both acute and chronic nephrotoxicity and triggers serious side effects. Although monitoring of tacrolimus exposure relies on clinical practice for determining trough levels, there is no agreement on the target levels during the first year in renal transplant recipients. While the largest clinical trial supported the minimization of tacrolimus exposure [20], one randomized clinical trial has shown that in case of steroid discontinuation and MMF reduction, maintaining TAC-C_0_ > 7 ng/mL after the fourth month reduces the risk of acute rejection and appearance of de novo DSAs without increasing renal toxicity [15]. Similarly, in low-immunological-risk renal transplants treated with TAC, reduced MMF, and low-dose steroids, TAC-C_0_ levels are associated with subclinical inflammation in patients monitored by protocol biopsies [16]. Additionally, it has been described that the effect of tacrolimus trough levels was modulated by the recipient’s baseline alloimmune risk, as defined by their class II HLA donor–recipient eplet mismatch [18].

In the present study, we analyzed whether interstitial inflammation is associated with clinical characteristics of donors and recipients as well as with transplant-related variables. In our cohort, the presence of interstitial inflammation was associated with lower TAC-C_0_ at the time of biopsy and with longer cold ischemia time, but it was not associated with mid-term renal function deterioration or the development of de novo DSA. Importantly, in our cohort, as in several others [6,16], few cases met the criteria for subclinical borderline rejection (11.7%) or TCMR/ABMR (5.8%). Since the presence of interstitial inflammation (i > 0) in otherwise normal biopsies has been associated with 15-year death-censored graft survival [6] in a similar way to SCR, we chose this threshold for our analysis. Notably, in other studies including patients treated with a steroid-free regimen, the incidences of borderline rejection and TCMR were significantly more frequent (31% and 20.8%) [21]. In this study, the authors did not find associations between TAC-C_0_ and subclinical inflammation, but it should be noted that at the time of the 3-month protocol biopsy, the TAC-C_0_ average was close to 10 ng/mL [22]. In this study, SCR within the first post-transplant year is associated with a significantly greater hazard of subsequent clinical rejection and death-censored graft loss. On the contrary, other studies have shown that T cell-mediated inflammation detected in protocol biopsies mostly reflects the injury–repair response to implantation stresses and has little relationship with future events and outcomes [23]. Acute kidney injury (AKI) after renal transplantation can also induce interstitial infiltration and tubulitis [24] leading to a histological picture indistinguishable from that of TCMR. In this sense, in our cohort of protocol biopsies, we observed an association between interstitial inflammation and longer cold ischemia time. Thus, the presence of interstitial inflammation is uncommon in our cohort of low-immunological-risk kidney transplants maintaining steroids (27.7%) and it is associated with tacrolimus exposure and cold ischemia time, suggesting that both immune and non-immune factors may contribute to subclinical inflammation in well-functioning grafts.

The disagreement between different studies on the prevalence of subclinical inflammation and its association with later clinical outcomes is partly explained by the inclusion of different populations and different maintenance immunosuppression regimens. However, there is general agreement that conventional biopsy assessment is limited due to poor interobserver reproducibility of individual lesions [11,25]. To overcome these limitations, it has been proposed that molecular phenotyping be incorporated. The application of microarrays to transplant biopsies has been an ongoing effort by many groups and the interpretation of molecular changes aided by the understanding of their biological mechanisms led to the grouping of different transcripts [10,14,26,27]. To summarize information derived from RNA microarrays, which evaluate thousands of genes, in the last reports of the Banff meetings gene sets containing a few hundred genes related to TCMR, ABMR, tissue-repair injury and other pathways leading to graft dysfunction were described [6,26,28,29]. In the present study, we evaluated the panel of genes described in the Banff meeting in 2017 via RT-PCR [14]. As expected, we confirm the discrimination capacity of a high number of these genes (111 out of 308 evaluated genes) to differentiate normal protocol biopsies from biopsies for cause with rejection. In the principal component analysis, we observed that TCMR-selective genes expressed in activated effector T cells (ADAMDEC1) and genes encoding different cytokines and their receptors mainly related to TCMR (CCL5, CXCL13 and CXLC9) were the most relevant in component 1, while genes encoding matrix proteins and solute transporters (COL1A1, SCL22A2 and SCL4A1) were the most relevant in component 2.

Regarding the derived gene set, we were interested in evaluating its relationship with tacrolimus exposure at the time of biopsy. We found that 19 of these 111 genes, mainly related to TCMR, were mildly correlated with TAC-C_0_, suggesting that a higher tacrolimus exposure contributes to a better control of subclinical inflammation. Interestingly, in the small set of normal protocol biopsies, we observed a close correlation between TAC-C_0_ and the expression of 2 out of these 19 genes (IKZF3 and CD2, Figure 4) IKZF3, expressed mostly in the lymph and spleen, is found in several immune cell types, including B cells, NK cells, CD4+, and CD8+ T cells. It is expressed most strongly in B cells and studies of IKZF3 knockout mice indicate a critical role for IKZF3 in B-cell differentiation, maturation, proliferation, and T cell-dependent B-cell responses. IKZF3 is upregulated in pre-B cells, and it has been found to play a role in executing the transition from large pre-B cells to small pre-B cells during normal B-cell development. IKZF3 has also been found to play an important role in T cell regulation. It is expressed in interleukin-17-producing helper T cells and promotes differentiation through silencing of interleukin 2 production. Recently, it has been shown that IKZF3 is upregulated not only in ABMR but also in TCMR urinary cell specimens suggesting that B cells may play a more active role in TCMR than previously recognized, perhaps functioning as classical antigen-presenting cells [30,31,32]. The CD2 family of costimulatory and adhesion molecules has also been shown to play a significant role in the execution of an alloimmune response since it is constitutively expressed by all T cells and upregulated upon antigen recognition. Importantly, CD2 is more highly expressed on effector memory T cells relative to central memory T cells and therefore more effectively targets those cells that are poised to rapidly exert effector function upon encounter with cognate antigen. In addition to its role in facilitating the adhesion of T cells to antigen-presenting cells during the immunological synapse, CD2 ligation results in the direct transmission of co-stimulatory signals to promote T cell activation and differentiation [33,34]. Importantly, in the evaluated set of biopsies, the expression of these 19 genes associated with TAC-C_0_ and split our protocol biopsy group into two clusters one containing all but one normal protocol biopsies and the other containing all rejection biopsies. The large set of protocol biopsies was distributed in a similar proportion in both clusters. Patients with protocol biopsies grouped in cluster 2 received a lower exposure to tacrolimus, showed more frequently an inflammatory phenotype, and displayed a faster decline of renal function in the mid-term. Thus, our results suggest that more adjusted immunosuppression during the early months after transplantation favors a better control of the inflammatory response and better preserving renal function in the mid-term.

Our effort to detect associations between gene expression, tacrolimus exposure, and HLA compatibility at the allelic or molecular level did not show significant associations in the multivariate analysis. It should be remarked that HLA typing in this cohort was performed according to clinical practice and thus, high-resolution HLA typing was not performed and the availability of HLA typing for all loci (especially DQ) was limited. However, in this cohort of successfully immunosuppressed renal transplant recipients, the number of patients developing de novo DSA was very low (2.9%) and although patients from cluster 2 tended to develop de novo DSA more frequently, this association did not reach statistical significance. Additionally, the present study has other important limitations, since associations between tacrolimus exposure and histological findings or gene transcripts were based on a single determination of TAC-C_0_ on the day of biopsy and a more refined evaluation of tacrolimus pharmacokinetics (e.g., area under the time–concentration curve) or pharmacodynamics (e.g., calcineurin activity) was not done. Finally, patients in group II (acute rejection) underwent biopsies later in comparison to patients in the other groups (protocol biopsies), and the impact of biopsy timing on gene expression has been widely acknowledged [11,35,36].

## 4. Materials and Methods

### 4.1. Patients

We considered renal transplants included in a prospective, observational study with an early (at 3–5 months) protocol biopsy performed between 2012 and 2019 as previously described [16]. Surveillance biopsies were performed in patients fulfilling the following criteria: (a) serum creatinine lower than 2 mg/dL; (b) stable renal function defined as a variability of serum creatinine lower than 15% between the determination at the time of biopsy and the previous one; (c) urinary protein creatinine ratio lower than 1 g/g; (d) non-use of oral anticoagulants; (e) non-technical difficulties to perform a renal biopsy (e.g., patients with large abdominal obesity, patients with large perirenal hematomas or patients with an idiomatic barrier were not considered) and (f) written informed consent. Two control groups of biopsies were selected from our biobank: strictly normal protocol biopsies (group I, n = 17) and biopsies for cause with either TCMR or ABMR (group II, n = 12) to generate a set of genes associated with rejection. Later, we selected a large sample of protocol biopsies (group III, n = 142) to evaluate whether gene expression was associated with donor and recipient characteristics or the intensity of immunosuppression. The flowchart of the study is shown in Figure 5.

Demographic characteristics of donors and recipients as well as transplant-related variables were recorded. Patients were followed in the outpatient area and the decline in renal function was estimated from the linear regression of all available measurements and expressed as mL/min/1.73 m^2^/year to adjust for the different timings of follow-up.

The present study has been approved by our Ethics Committee (Comité Etico de Investigación Clínica del Hospital Universitari Vall d’Hebron PR(AG)369/2014, approval date 1 December 2014) and all participants signed written informed consent. The study was conducted by the Declaration of Helsinki and adhered to the Principles of the Declaration of Istanbul on Organ Trafficking and Transplant Tourism.

### 4.2. HLA Typing and HLA Antibodies

The recipients’ and donors’ HLA typing was performed by DNA-based low-resolution typing with sequence-specific primers (SSP). For class I (A and B loci) and for class II (DR loci), results were available for all donor/recipient pairs. HLA C typing was available in 116 cases from donors and 47 cases for recipients. HLA DQ loci typing was available for 40 donors and 26 recipients.

The HLA Matchmaker program (Rene Duquesnoy, 2016, University of Pittsburgh Medical Center, Pittsburgh, PA HLA-ABC Eplet Matching Version 3.1 and DRDQDP Eplet Matching Program V3.1 from http://www.epitopes.net/downloads.html, (accessed on 17 November 2022) was used to calculate eplet scores. Donor and recipient typing was converted to high resolution using a local frequency table typed by sequence-based typing. Total numbers of incompatible eplets and antibody-verified eplets were calculated.

PIRCHE-II scores were calculated using version 3.3 from https://www.pirche.org (accessed on 22 May 2023). The NetMHCIIpan 3.0 algorithm predicted nonameric-binding cores of donor mismatched HLA-derived peptides that could bind to recipient HLA-DRB1. For cases with only low-resolution HLA typing, PIRCHE-II generates a potential high-resolution HLA typing and PIRCHE-II was calculated for each potential typing for both donors and recipients. These values were weighted by haplotype frequencies in the general population as validated in a previous study [37].

Anti-HLA antibodies on the day of transplant, biopsy, and during follow-up were determined using a single-antigen class-I and class-II flow beads-assay kit (LIFECODES, division of Immucor, Stanford, CA, USA). Beads with a normalized MFI > 500 were considered positive if (MFI/MFI lowest bead) > 5.

### 4.3. Immunosuppression

Induction and maintenance of immunosuppression with tacrolimus, MMF, and steroids were performed as previously described [16]. Target TAC-C_0_ levels during the first 3 months were 8–12 ng/mL and 6–10 ng/mL thereafter. Exposure to tacrolimus was evaluated using a concentration dose ratio (C/D), coefficient of variation of TAC-C_0_ until the day of biopsy, and TAC-C_0_ at the time and time in/above/below therapeutic range of biopsy, as previously described [38].

### 4.4. Biopsies

Ultrasound-guided renal biopsies were performed with a 16G automated needle, and 3 cores of tissue were obtained: one was processed for optical microscopy; one was embedded in OCT for immunofluorescence and the other one was stored in RNA later. Histological lesions were evaluated according to the last Banff criteria [9] and the definition of borderline changes were foci of tubulitis (t1–t3) with mild interstitial inflammation (i1) or mild tubulitis (t1) with moderate–severe interstitial inflammation (i2–i3). C4d was stained with indirect immunofluorescence with a monoclonal antibody (Quidel, San Diego, CA, USA) and deposition in peritubular capillaries was graded according to the Banff criteria. The third core was stored with Ambion^®^ RNAlater^®^ Tissue Collection at −80 °C (Applied Biosystems, Austin, TX, USA).

### 4.5. Analysis Using Fluidigm Microfluidics Dynamic Arrays

Total RNA extraction, assessment of RNA quality, and cDNA synthesis were done as previously described [13]. The aim of the study was the quantitative analysis of 318 genes (308 target genes and 10 housekeeping genes: ECD, EIF1, FUBP3, GGNBP2, GNB1, RPN1, RPN2, SERBP1, UBC, UBE2D3) in the biopsies. The 308 target genes were selected from a list of identified, non-repeated prime gene lists reported in the Banff 2017 meeting [14]. For this purpose, we used the Biomark HD Nanofluidic Quantitative PCR (qPCR) system (Fluidigm Corporation, San Francisco, CA, USA) combined with GE 96.96 Dynamic Arrays IFCs. For sequence detection, predesigned Primetime qPCR primer assays or custom primer assays were used for amplification and detection using the EvaGreen fluorochrome. The assays have been divided into four IDT assay plates and all housekeeping genes have been included in all plates. Samples were treated with Exonuclease I (Exo I) (Thermo Scientific EN0582, Willmington, DE, USA) to remove unincorporated primers. QIAGEN^®^ Multiplex PCR Kit Cat N.206143 (Hilden, Germany) was used for the specific target amplification. According to the manufacturer’s instructions, 13 genes were eliminated due to potential amplification of genomic DNA and 12 genes were also not considered due to lack of expression in more than half of the samples. The analysis of the expression of the cDNA was performed with Biomark HD Nanofluidic qPCR system (Fluidigm Corporation, San Francisco, CA, USA) combined with 96.96 Dynamic Arrays IFCs by employing the Master Mix Sso FastTM Eva Green^®^ Supermix with Low ROX (Bio-Rad Laboratories, Hercules, CA, USA). The Ct (Cycle Threshold) data and the Quality Call of the amplification curve were determined by the Fluidigm Real-Time PCR Analysis Software version 4.1.3. Samples with a Ct value higher than 27 (n = 5) were eliminated since they are not reliable according to Fluidigm, the owner of the technology (https://www.fluidigm.com, accessed on 21 December 2020). All procedures were conducted as part of the genomics and proteomics service of the Universidad del País Vasco Science Park (Centro de Biotecnología María Goyri).

### 4.6. Statistics

Results are expressed as raw numbers for categorical variables, as the mean ± standard deviation for continuous normally distributed variables and median (interquartile range) for non-normally distributed variables. To compare unpaired data, Fisher’s exact test, Mann–Whitney U test, Student’s *t*-test, Kruskal–Wallis, and analysis of variance were applied according to the distribution of variables. Logistic regression analysis was employed for multivariate analysis. Kaplan–Meier analysis was employed for survival analysis with a log-rank test for comparisons between groups. All *p*-values were two-tailed and a *p*-value < 0.05 was considered significant.

### 4.7. Bioinformatic Analysis

Bioinformatic analysis was performed at the Statistics and Bioinformatics Unit (UEB) of the Vall d’Hebron Institute of Research (VHIR, Barcelona, Spain). The analyses were carried out with the statistical program “R” (R version 3.6.3 (), Copyright (C) 2021 The R Foundation for Statistical Computing, https://www.R-project.org/, accessed on 29 February 2020). A comprehensive quality control process was applied to assess the suitability of all samples for inclusion in the study. The calculation of relative quantification (RQ = 2^−∆∆Ct^) was performed according to Livak’s method [39]. A principal component analysis (PCA) was performed to describe how the samples are grouped according to the Ct values obtained. Because the variability between genes used as normalizers was low, all were used as housekeeping genes. The geometric mean of the Ct values of the housekeeping genes was obtained as described by Vandesompele et al. [40]. In the process of normalization, the Ct values of each gene were subtracted from the geometric mean value of the two housekeeping genes selected to obtain the ∆Ct values. Later, they were used to make comparisons. Spearman’s correlation between the expression of each of the genes and tacrolimus levels was performed to select significant genes following criteria of fold change (FC) and statistical significance (FC > 3 and *p*-value < 0.01).

## 5. Conclusions

In summary, we evaluated a cohort of patients with an early protocol biopsy and observed that lower tacrolimus through the level at the time of biopsy was associated with interstitial inflammation and a higher expression of rejection-associated transcripts in stable grafts. Cluster analysis allowed the detection of a group of patients who had lower tacrolimus through levels at the time of biopsy, who showed an inflammatory phenotype and displayed a faster decline in renal function in the mid-term. Thus, our results suggest that adjusted immunosuppression during the early months after transplantation favors better control of the inflammatory response without deleterious effects on renal function in the mid-term. Furthermore, although transcriptomic analysis is not currently widely available in most renal transplant units, its future integration into clinical practice could contribute to improving the management of immunosuppression in renal transplant recipients.

## Figures and Tables

**Figure 1 ijms-25-03189-f001:**
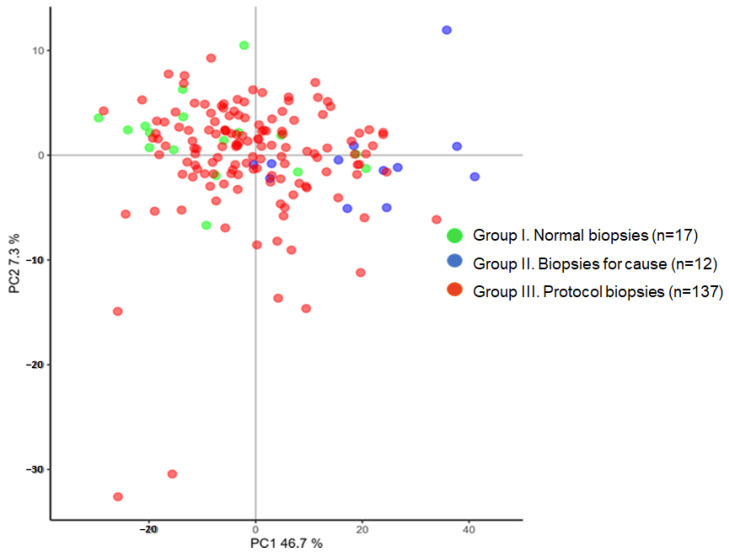
Principal component analysis with the expression of 307 genes in the different study groups.

**Figure 2 ijms-25-03189-f002:**
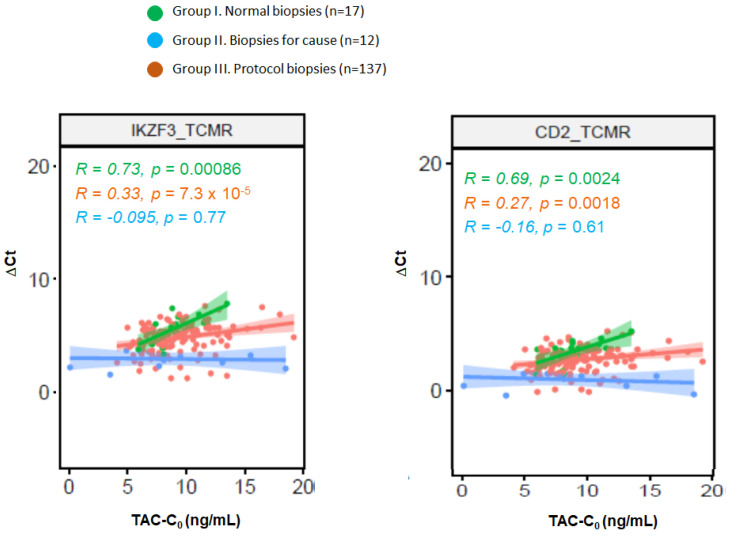
Correlation between tacrolimus trough levels (TAC-C_0_) at the time of biopsy and the expression of IKZF3 and CD2 genes in the different study groups.

**Figure 3 ijms-25-03189-f003:**
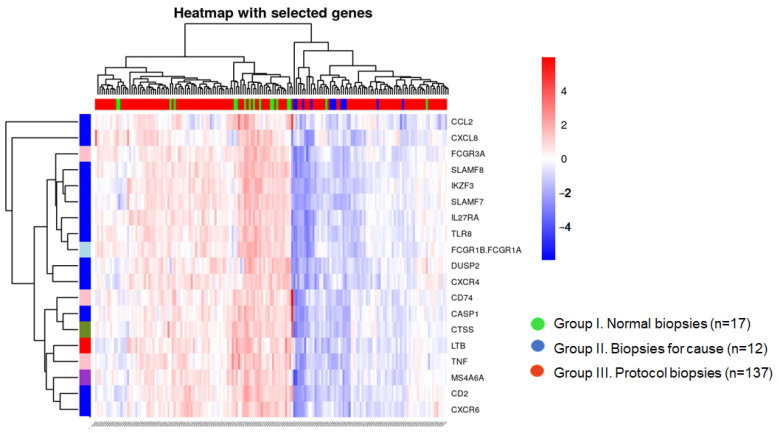
Heatmap of the 19 genes correlated with tacrolimus trough levels (TAC-C_0_) at the time of biopsy in the different study groups.

**Figure 4 ijms-25-03189-f004:**
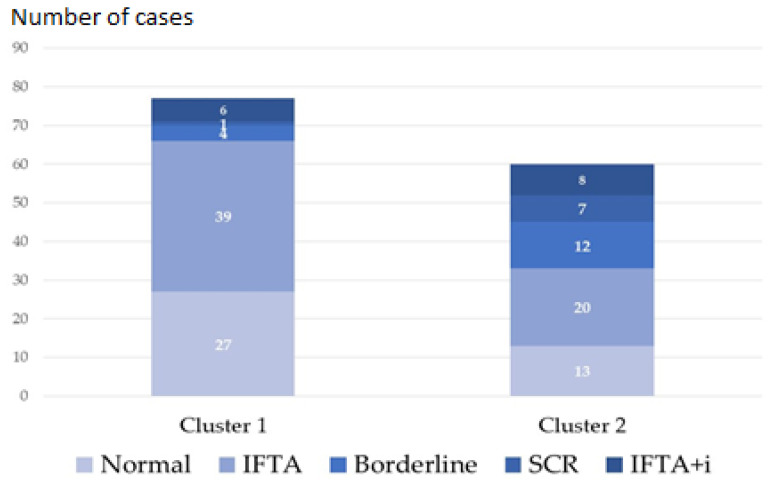
Histological diagnosis in the two clusters of biopsies was defined according to the gene expression of the 19 genes correlated with tacrolimus trough levels (TAC-C_0_) at the time of biopsy.

**Figure 5 ijms-25-03189-f005:**
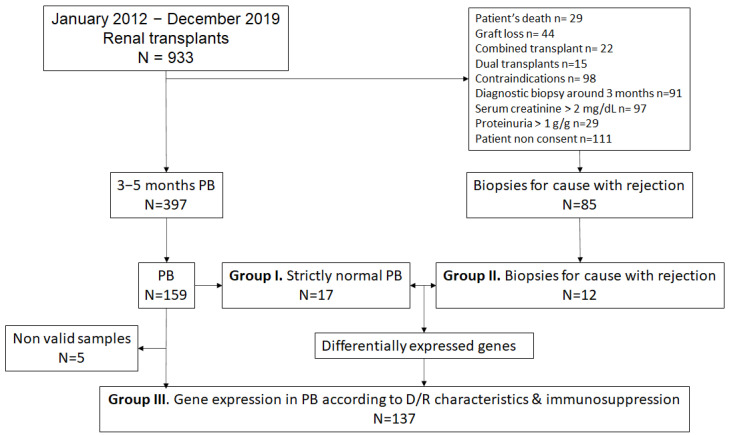
Flow chart of the patients and biopsies included in the study. PB—protocol biopsies.

**Table 1 ijms-25-03189-t001:** Donor and recipient characteristics as well as transplant-related variables in the 3 study groups.

Variable	Group I (n = 17)	Group II (n = 12)	Group III (n = 137)
Donor type (BDD/DCD/LD)	10/4/3	7/2/3	85/34/18
Donor age (years)	45 ± 18	59 ± 15	57 ± 14
Donor gender (m/f)	10/7	4/8	77/60
Recipient age (years)	46 ± 13	50 ± 16	55 ± 14
Recipient gender (m/f)	10/7	6/6	91/46
First transplant/retransplant	15/2	7/5	117/20
Primary renal disease (GN/ADPKD/diabetes/others/unknown)	3/5/0/4/5	4/1/1/2/4	28/18/11/25/55
Class I HLA mismatch (A + B)	2.5 ± 0.9	2.1 ± 0.9	2.8 ± 1.0
Class II HLA mismatch (DR)	1.1 ± 0.5	1.3 ± 0.6	1.1 ± 0.6
Induction (basiliximab/thymoglobulin)	8/9	4/8	77/60
Cold ischemia time	14.3 ± 6.7	13.0 ± 7.0	13.4 ± 6.8
Delayed graft function (no/yes)	16/1	10/2	123/19
Previous episodes of rejection (no/yes)	17/0	9/3	134/7
DSA at the time of transplant (no/yes)	15/2	11/1	130/7
CMV infection (no/viremia/disease)	14/2/1	9/3/0	115/18/4

Group I—normal protocol biopsies; Group II—biopsies with rejection; group III—protocol biopsies with different histological phenotypes BDD—brain death donor; DCD—donation after circulatory death; LD—living donor; GN—glomerulonephritis; ADPKD—autosomal dominant polycystic kidney disease; DSA—HLA donor-specific antibodies; CMV—cytomegalovirus. Mean ± SD or raw numbers are employed to describe variables.

**Table 2 ijms-25-03189-t002:** Data at the time of biopsy in the 3 study groups.

Variable	Group I (n = 17)	Group II (n = 12)	Group III (n = 137)
Time of biopsy (months)	4.7 ± 1.7	43 ± 55	4.4 ± 1.4
Serum creatinine (mg/dL)	1.22 ± 0.31	2.72 ± 1.95	1.44 ± 0.32
eGFR (mL/min/1.73 sqm)	66.6 ± 23.0	35.1 ± 21.1	52.2 ± 14.6
Urine P/C ratio (mg/g)	260 ± 190	1890 ± 1340	265 ± 192
DSA at the time of biopsy (no/yes)	17/0	8/4	133/4
Tacrolimus dose (mg/day)	6.8 ± 4.2	7.6 ± 5.9	6.5 ± 4.2
TAC-C_0_ (ng/mL)	8.8 ± 2.0	7.9 ± 3.6	9.3 ± 2.7
MMF dose (g/day)	1.0 ± 0.2	0.9 ± 0.2	0.9 ± 0.2

Group I—normal protocol biopsies; Group II—biopsies with rejection; Group III—protocol biopsies with different histological phenotypes; eGFR—estimated glomerular filtration rate according to the CKD-EPI formula; urine P/C ratio; protein to creatinine ratio in a spot morning urine sample; DSA—HLA donor-specific antibodies; TAC-C_0_—tacrolimus trough levels; MMF—mycophenolate mofetil. Mean ± SD or raw numbers are employed to describe variables.

**Table 3 ijms-25-03189-t003:** Donor/recipient characteristics and transplant-related variables according to interstitial inflammation in the protocol biopsy.

Variable	i-Score = 0 (n= 99)	i-Score ≥1 (n = 38)	*p*-Value
Donor type (BDD/DCD/LD)	58/27/14	27/7/4	0.490
Donor age (years)	57 ± 14	56 ± 15	0.837
Donor gender (m/f)	56/43	21/17	0.793
Recipient age (years)	55 ± 14	57 ± 14	0.453
Recipient gender (m/f)	65/34	26/12	0.759
First transplant/retransplant	85/14	32/6	0.807
Class I HLA mismatch (A + B)	2.8 ± 0.9	2.9 ± 1.0	0.521
Class II HLA mismatch (DR)	1.1 ± 0.6	1.2 ± 0.6	0.233
Class I Eplet mismatch	14 ± 6	14 ± 8	0.845
Class II Eplet mismatch	15 ± 10	17 ± 15	0.305
PIRCHE-II class I	49 ± 27	49 ± 29	0.914
PIRCHE-II class II	34 ± 25	34 ± 22	0.996
DSA at the time of transplant (no/yes)	96/3	34/4	0.074
Induction (Basiliximab/ATG)	53/46	24/14	0.367
Cold ischemia time	12.6 ± 6.9	15.3 ± 6.2	0.040
DGF (no/yes)	87/12	31/7	0.339
TCMR before protocol biopsy (no/yes)	94/5	36/2	0.960

BDD—brain death donor; DCD—donation after circulatory death; LD—living donor; DGF—delayed graft function; DSA—donor-specific antibodies. Mean ± SD or raw numbers are employed to describe variables.

**Table 4 ijms-25-03189-t004:** Data at the time of biopsy according to interstitial inflammation in the protocol biopsy.

Variable	i-Score = 0 (n= 99)	i-Score ≥1 (n = 38)	*p*-Value
Time of biopsy (months)	4.3 ± 1.4	4.6 ± 1.7	0.169
Serum creatinine (mg/dL)	1.5 ± 0.3	1.4 ± 0.3	0.747
eGFR (mL/min/1.73 m^2^)	52 ± 14	53 ± 16	0.790
Urine P/C ratio (mg/g)	275 ± 206	239 ± 148	0.331
DSA at the time of biopsy (no/yes)	96/3	37/1	0.901
Tacrolimus dose (mg/day)	6.7 ± 4.5	6.0 ± 3.3	0.425
TAC-C_0_ (ng/mL)	9.7 ± 2.7	8.2 ± 2.2	0.002
C/D tacrolimus (ng/mL/mg)	1.72 (1.06–2.80)	1.54 (1.05–1.98)	0.220
CV TAC- C_0_ from day 7 to biopsy (%)	36.5 ± 23.3	36.6 ± 14.3	0.990
Time in TR (%)	67 ± 31	69 ± 33	0.738
Time above TR (%)	24 ± 28	17± 26	0.184
Time below TR (%)	7 ± 14	10 ± 18	0.303
MMF dose (g/day)	0.9 ± 0.2	0.9 ± 0.2	0.735
eGFR decline (mL/min/1.73 m^2^/year)	−0.8 ± 4.3	−1.1 ± 2.9	0.163

eGFR—estimated glomerular filtration rate by MDRD-4 formula; urine P/C ratio—urine protein creatinine ratio; TAC-C_0_—tacrolimus trough levels at the time of biopsy; CV of TAC-C_0_—coefficient of variability of tacrolimus; C/D—concentration dose ratio of tacrolimus. Mean ± SD or raw numbers are employed to describe variables.

**Table 5 ijms-25-03189-t005:** Donor/recipient characteristics, transplant-related variables, and data at the time of biopsy in patients from both clusters according to gene expression.

Variable	Cluster 1 (n = 77)	Cluster 2 (n = 60)	*p*-Value
Donor type (BDD/DACD/LD)	47/17/13	38/17/5	ns
Donor age (y)	54 ± 13	60 ± 15	0.0316
Patient age (y)	54 ± 13	57 ± 15	ns
Patient sex (m/f)	53/24	38/22	ns
First transplant/retransplant	68/9	49/11	ns
Class I HLA mismatch (A + B)	2.7 ± 1.0	3.0 ± 1.0	0.039
Class II HLA mismatch (DR)	1.1 ± 0.7	1.2 ± 0.6	0.571
HLA eplet class I mismatch	13 ± 6	15 ± 7	0.061
HLA eplet class II mismatch	14 ± 11	16 ± 9	0.335
HLA AbV eplet DRB mismatch	2.8 ± 2.4	3.6 ±2.4	0.059
HLA AbV eplet DQB mismatch	2.6 ± 2.6	2.5 ± 2.3	0.757
PIRCHE-II class I	48 ± 28	52 ± 28	0.446
PIRCHE-II class II	34 ±28	34 ± 19	0.999
Induction (basiliximab/thymoglobulin)	44/33	33/27	ns
DGF (n/y)	67/10	51/9	ns
TCMR before protocol biopsy (n/y)	72/5	68/2	ns
eGFR (mL/min/1.73 sqm) biopsy	53 ± 13	51 ± 16	ns
Urinary protein/creatinine (g/g) biopsy	0.24 ± 0.17	0.30 ± 0.24	ns
TAC-C_0_ (ng/mL) biopsy	9.8 ± 2.6	8.6 ± 2.6	0.0133
CV of TAC-C_0_ from day 7 to biopsy (%)	34.9 ± 22.4	39.2 ± 20.5	0.2483
Time in TR (%)	68 ± 32	50 ± 30	0.430
Time above TR (%)	25 ± 30	15 ± 25	0.304
Time below TR (%)	5 ± 15	12 ± 18	0.070
MMF dose (g/day)	0.9 ± 0.2	0.9 ± 0.2	ns
eGFR decline (mL/min/1.72 m^2^/year)	−0.2 ± 3.7	−1.9 ± 4.1	0.0145

BDD—brain death donor; DCD—donation after circulatory death; LD—living donor; DGF—delayed graft function; TCMR. T cell-mediated rejection; N—normal; BL—borderline lesions; SCR—subclinical rejection; IFTA—interstitial fibrosis—and tubular atrophy; eGFR—estimated glomerular filtration rate; TAC-C_0_—tacrolimus trough levels at the time of biopsy. Mean ± SD or raw numbers are employed to describe variables.

## Data Availability

Data are contained within the article and Appendix A.

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
