# Peer review of "Expression of Rejection-Associated Transcripts in Early Protocol Renal Transplant Biopsies Is Associated with Tacrolimus Exposure and Graft Outcome"

_ijms, 2024, doi:10.3390/ijms25063189_

Round 1
Reviewer 1 Report
Comments and Suggestions for Authors
The manuscript lacks clarity and readability
Author Response
Consulte el archivo adjunto.

Reviewer 2 Report
Comments and Suggestions for Authors
Dear editors:
It is a great honor and pleasure for me to be invited as the reviewer for this important work entitled “Expression of rejection-associated transcripts in early protocol renal transplant biopsies are associated with tacrolimus exposure and graft outcome”. Betty Chamoun and co-authors investigated the association between rejection-related transcripts in early protocol renal transplant biopsies, tacrolimus exposure and graft outcome. This study topic is novel and advanced, attributing to Prof. Francesc Moreso’s long-term efforts and contributions in this scientific field. I have a number of comments concerning this study:
1. The calcineurin inhibitor tacrolimus-induced nephrotoxicity is confirmed worldwide. In light of their conclusion “a more intense immunosuppression during the early months after transplantation favors a better control of the inflammatory response without deleterious effects on renal function on the mid-term”, it should be rephrased to highlight that using optimal dose to achieve the therapeutic target of immunosuppression is of great importance. Moreover, the authors should be cautious for sepsis due to intensive immunosuppression therapy.
2. From the perspective of a clinician, targets from transcript/biopsy results to guide the therapeutic dose of immunosuppression should be developed and discussed.
3. Table 2: The variation of “time of biopsy” is unusual.
4. All tables: The huge heterogeneity between groups was noted, resulting in potential bias.
5. Table 2: Tacrolimus dose is higher in group 2 than 1? The P-values should be provided appropriately in table 1and 2.
6. Table 3: Cold ischemia time is a critical issue to cause i-score ≥1.
7. Table 4: Why did Tacrolimus and MMF dose not differ between groups?
8. Too many typo errors limit the scientific merits of the study: Line 382: et => at
9. Table 5: Ns; Significant figures should be unified.
10. The abbreviation list should be provided in the text: TCMR or ABMR, etc.
The research is interesting that should be published after appropriate revision.
Comments on the Quality of English LanguageModerate editing of English languag is required.
Author Response
Consulte el archivo adjunto

Reviewer 3 Report
Comments and Suggestions for Authors
This paper examining transcripts for rejection and allograft outcome is of interest and mostly well written. I have a few comments:
1) Because of the layout of the paper its not clear early on what the 3 groups are- group 1 is not identified in patients or biopsies lines 83-90 although groups 2 and 3 are or in the table 1. I suggest that the authors make this more obvious- it was only by reading later in the paper that we know that group 1 were "normal" biopsies.
2) The numbers of normal 17 and abnormal (12) are relatively small in number compared with the group 3- how confident can we be that this is adequately robust to identify transcripts and correctly classify them- how were such numbers chosen and what evidence/power do the authors propose to reassure us they are sufficient?
3) The very small number of DSAs (n=4) render any formal analysis meaningless and trying to present KM plots of such and referring to this data in forms of claiming that the clusters for example show a trend to cluster 2 is not justified- I recommend this data is withdrawn in its current format and amended simply to the comment they already make in the discussion on line 224 that they cannot make any comment regarding DSAs given the small numbers.
Author Response
Consulte el archivo adjunto
